# Complete Mitochondrial Genome Sequence and Identification of a Candidate Gene Responsible for Cytoplasmic Male Sterility in Celery (*Apium graveolens* L.)

**DOI:** 10.3390/ijms22168584

**Published:** 2021-08-10

**Authors:** Qing Cheng, Peng Wang, Tiantian Li, Jinkui Liu, Yingxue Zhang, Yihao Wang, Liang Sun, Huolin Shen

**Affiliations:** 1Beijing Key Laboratory of Growth and Developmental Regulation for Protected Vegetable Crops, Department of Vegetable Science, College of Horticulture, China Agricultural University, Beijing 100193, China; chengqing2020@cau.edu.cn (Q.C.); LTT20140530@163.com (T.L.); 18435709400@163.com (J.L.); qwe2171305@126.com (Y.Z.); yhwang0906@126.com (Y.W.); liang_sun@cau.edu.cn (L.S.); 2Institute of Vegetable Research, Guangxi Academy of Agricultural Sciences, Nanning 530007, China; wangpeng@gxaas.net

**Keywords:** celery, cytoplasmic male sterility (CMS), mitochondrial genome, CMS-associated gene

## Abstract

Celery (*Apium graveolens* L.) is an important leafy vegetable worldwide. The development of F_1_ hybrids in celery is highly dependent on cytoplasmic male sterility (CMS) because emasculation is difficult. In this study, we first report a celery CMS, which was found in a high-generation inbred line population of the Chinese celery “tanzhixiangqin”. Comparative analysis, following sequencing and assembly of the complete mitochondrial genome sequences for this celery CMS line and its maintainer line, revealed that there are 21 unique regions in the celery CMS line and these unique regions contain 15 ORFs. Among these ORFs, only *orf768a* is a chimeric gene, consisting of 1497 bp sequences of the *cox1* gene and 810 bp unidentified sequences located in the unique region, and the predicted protein product of orf768a possesses 11 transmembrane domains. In summary, the results of this study indicate that *orf768a* is likely to be a strong candidate gene for CMS induction in celery. In addition, *orf768a* can be a co-segregate marker, which can be used to screen CMS in celery.

## 1. Introduction

Celery (*Apium graveolens* L., 2n = 2× = 22), an annual or biennial herbage species that belongs to the family Apiaceae, is one of the most important vegetable crops worldwide. Because celery has numerous small flowers and the maturation stage of male and female flowers is inconsistent, manual emasculation in celery is extremely difficult. Emasculation can be avoided using the male sterile line, which can significantly reduce the cost of commercial hybrid seed production in celery.

Male sterility can be classified into two types: genic male sterility (GMS) and cytoplasmic male sterility (CMS). Compared to GMS, CMS is more widely used because of the 100% maintenance ability of male sterility. To date, CMS has been successfully used in production of many commercial F_1_ seeds, such as rice, maize, Chinese cabbage, and pepper [1,2].

CMS is often found in higher plants and is a maternally inherited trait, in which the plants develop normally but with abnormal stamen and incompatible pollen. At present, it is generally believed that CMS is mainly caused by the rearrangement of the mitochondrial genome, which can result in novel open reading frame (ORF) genes [1]. These ORFs often have chimeric structures or are co-transcribed with known mitochondrial genes, and their encoded proteins often have transmembrane domains [3,4]. The accumulation of these novel chimeric ORFs at the RNA or protein level might lead to the loss of function of normal mitochondrial genes, ultimately leading to male sterility [4,5].

To date, many CMS-associated genes have been identified in different plant species [1,6]. In CMS-BT and CMS-HL rice, *orf79* and its variant gene *orfH79* have a chimeric structure with *cox1* [7,8]. In CMS-WA rice, *orf352* has been reported to be co-transcribed with the ribosomal protein gene *rpl5,* and its encoded protein has three transmembrane domains [4], and *orf162* has a chimeric structure with *rpl2* and transmembrane domains [9]. In radish, *orf463* and *orf463a* have been reported in CMS-DCGMS and CMS-NWB, respectively, both of which have a chimeric structure with *cox1* and possess 12 potential transmembrane domains, and these two genes are identical [10,11]. In *Brassica juncea*, *orf288* is located downstream of *atp6* and is co-transcribed with *atp6* in *hau* CMS, which is also a chimeric gene composed partially of *nad5* and *orf293* [12]. Furthermore, transgenic studies have shown that *orf288* is related to male sterility in *hau* CMS *Brassica juncea* [13]. In onion, *orf725* was identified as a strong candidate gene for both CMS-S and CMS-T cytoplasms, a chimeric gene consisting of almost the entire *cox1* gene and other sequences [14,15,16].

In most CMS cases, male sterility can be restored by nuclear restorer-of-fertility (*Rf*) genes. To date, many *Rf* genes have been cloned, and most of them are pentatricopeptide repeat (PPR) genes [1,17]. For instance, *Rf4* in CMS-WA rice, *Rf5* and *Rf6* in CMS-HL rice, *Rf1a* and *Rf1b* in CMS-BT rice [7,18,19,20,21], *Rfo* and *Rfk1* in Ogu-CMS and Kos-CMS radish [22,23,24], *Rfp* in CMS-Pol oilseed rape [25] are PPR genes. However, there are also some *Rf* genes that are not PPR genes, such as *Rf2* in T-cytoplasm maize, which encodes an aldehyde dehydrogenase [26], *Rf2* in CMS-LD rice that encodes a mitochondrial glycine-rich protein [27] and *Rf17* in CMS-CW rice, which encodes a mitochondrial sorting protein containing an acyl-carrier protein synthase-like domain [28].

In this study, we first report the assembly of the complete mitochondrial genome sequences for celery. Comparative analysis of the mitochondrial genomes of the celery CMS line W99A and its maintainer line W99B confirmed that *orf768a* was a strong candidate for the cytoplasmic male sterility gene in W99A. In addition, our sequence data may contribute to understanding the evolution of the Umbelliferae mitochondrial genome.

## 2. Results

### 2.1. Flower Morphology of Celery CMS Line and Maintainer Line

There was no significant difference in vegetative growth between the celery CMS and maintainer lines, except when they bloom. In the maintainer line, flowers were normal, filaments were elongated, anthers cracked normally, and the pollen dispersed on the surface (Figure 1a,b). The celery CMS line can be divided into two types: one type is characterized by the complete degeneration of the stamen, showing no filaments and anthers (Figure 1c,d); the other type is characterized by the incomplete degeneration of the stamen, showing shortened filaments, withered anther that was not dehiscent, and no pollen dispersal (Figure 1e,f). Both of these two CMS lines produced almost no pollen grains (Figure 1g), whereas the maintainer line produced normal pollen grains (Figure 1h). The celery CMS line W99A used in this study belonged to the incomplete stamen degeneration type.

### 2.2. Microspore Development of Celery CMS Line and Maintainer Line

To determine the period of anther defects in the celery CMS line W99A, paraffin sectioning of anthers from this line and its maintainer line W99B was performed. At the stamen primordium stage, sporogonia, sporogenetic cell stage, and pollen mother cell stage, there were no obvious significant differences in microspore development between the celery CMS line W99A and its maintainer line W99B (Figure 2a–c,g–i). However, the tapetum cells of CMS line W99A were highly vacuolated at the tetrad stage (Figure 2d), then it became over-vacuolated and squeezed the microspores toward the centre at the uninucleate stage (Figure 2e), finally inhibiting callosum degradation and microspore release at the mature pollen stage (Figure 2f), whereas, in the maintainer line W99B, normal tetrads, tapetum development, and fertile pollen grains were observed (Figure 2j–l).

### 2.3. Mitochondrial Genome Sequence and Identification ORFs in CMS Line W99A and Its Maintainer Line W99B

The mitochondrial genomes of the celery CMS line W99A and its maintainer line W99B were assembled as 371,275 bp and 394,073 bp circular molecules, respectively (Figure 3), with GC contents of 45.09% and 45.05%, respectively. The W99A and W99B assembled mitochondrial genomes were then searched for ORFs that encoded ≥100 amino acids. In the W99A mitochondrial genome, 172 ORFs and 35 non-coding RNAs (ncRNAs) were identified, while in the W99B mitochondrial genome, there were 178 ORFs and 41 non-coding RNAs. Among the above ORFs, there were 41 known genes and 44 known genes in the W99A and W99B mitochondrial genomes, respectively (Appendix A). Among these known genes in W99A mitochondrial genome, there were eight encoding ribosomal protein large subunits *(rpl5-1/2*, *rpl10*, *rpl16-1/2*, *rps1*, *rps3-1/2*, *rps7*, *rps12-1/2,* and *rps13-1/2*), four ATP synthase (*atp1*, *atp6*, *atp8,* and *atp9-1/2*), three cytochrome oxidase subunits (*cox1*, *cox2*, *cox3*), one cytochrome b (*cob*), nine NADH dehydrogenase subunits (*nad1*, *nad2*, *nad3-1/2*, *nad4-1/2*, *nad4L*, *nad5*, *nad6*, *nad7-1/2,* and *nad9*), four cytochrome C synthesis related proteins (*ccmB*, *ccmC-1/2*, *ccmFc,* and *ccmFN*), one mature enzyme (*matR*), and one mitochondrial membrane transport protein (*mttB*). Whereas in W99B mitochondrial genome, there were eight encoding ribosomal protein large subunits (*rpl5*, *rpl10*, *rpl16*, *rps1-1/2*, *rps*3, *rps7-1/2*, *rps12,* and *rps13*), four ATP synthase (*atp1-1/2/3*, *atp6*, *atp8,* and *atp9*), three cytochrome oxidase subunits (*cox1-1/2*, *cox2,* and *cox3*), one cytochrome b (*cob-1/2*), nine NADH dehydrogenase subunits (*nad1*, *nad2-1/2*, *nad3*, *nad4-1/2*, *nad4L-1/2*, *nad5*, *nad6*, *nad7*, and *nad9*), four cytochrome C synthesis related proteins (*ccmB-1/2/3*, *ccmC*, *ccmFc*, and *ccmFN-1/2*), one mature enzyme (*matR*), and one mitochondrial membrane transport protein (*mttB-1/2*) (Appendix A). There were 21 and 20 unique ORFs in W99A and W99B, respectively (Appendix A).

### 2.4. Repeat Sequence Analysis of W99A and W99B Mitochondrial Genomes

As shown in Appendix A, both of the celery CMS line W99A and its maintainer line W99B has 117 repeats, and the proportion of total repeat sequence in W99A and i W99B was 65.03% and 72.14%, respectively. The largest repeats in W99A and W99B was 31,917 bp and 71,571 bp, respectively. And the large repeats (more than 1000 bp) of W99A and W99B take the majority of the whole repeats, which accounted for 96.95% and 97.44%, respectively. These results indicated that the repeats, especially the large repeats, contributed greatly to the large mitochondrial genome size of celery.

### 2.5. Comparative Analysis of W99A and W99B Mitochondrial Genomes

A total of 14 syntenic sequence blocks were identified in the mitochondrial genomes of W99A and W99B (Figure 3), accounting for 61.56% and 57.97% of the mitochondrial genome sequences, respectively. In addition, 47 unique regions ranging from 23 bp to 9898 bp were found between W99A and W99B, with 21 unique regions in the W99A genome and 26 unique regions in the W99B genome (Appendix A).

To clarify the difference between W99A and W99B mitochondrial genomes, we conducted genome structural variations analysis by setting W99A as a reference. As shown in Figure 4, there were 23 ectopic, 5 inversions, and 21 ectopic + inversion regions between the W99A and W99B mitochondrial genomes. Meanwhile, four insertions and one deletion longer than 50 bp were identified in the syntenic regions.

### 2.6. SNP and InDel Detection in W99A and W99B Mitochondrial Genomes

To identify sequence variations in the predicted genes between the W99A and W99B mitochondrial genomes, we searched for SNPs and InDels in the mitochondrial genomes of W99A and W99B. As shown in Appendix A, 69 SNPs and 9 InDels were identified between the W99A and W99B mitochondrial genomes. Among those SNPs and InDels, nine SNPs caused non-synonymous mutations, one SNP caused premature stop (Appendix A), and three InDels caused frame-shifted mutations (Appendix A). These mutated ORFs included the two known mitochondrial genes *cob* and *rps1*, which have no relationship with CMS (Appendix A). The ORFs encoding proteins with length < 100 amino acids were not considered to be CMS candidate genes, and the other ORFs (*orf165a*, *orf174a*, *orf182a*, *orf192a-2*) did not have a chimeric structure. Therefore, these mutated ORFs and the two known mitochondrial genes *cob* and *rps1* were not considered to be CMS candidate genes.

### 2.7. Selection of Candidate Genes That Are Responsible for CMS Celery

Based on previous studies, novel ORFs located in the unique regions of CMS lines are likely to be the genes that determine the CMS trait. In our study, 21 specific ORFs (encoding proteins with ≥100 amino acids) were identified in W99A mitochondrial genome (Appendix A). Among these ORFs, there were 15 ORFs in which full-length or partial sequences were detected in the unique regions of W99A (Appendix A).

It is considered that most of the known or strong candidate CMS genes are chimeric in structure, co-transcribed with functional genes, or encode a peptide with transmembrane domains. Most of the W99A specific ORFs were neither chimeric in structure nor co-transcribed with functional genes, except orf—*orf768a*, with a 5′ end containing the almost complete sequence of the *cox1* gene and a 3′ end with unidentified sequences (Figure 5a). Seven ORFs (*orf115b*, *orf163a*, *orf241a*, *orf254a*, *orf340a-1/2*, *orf402a-1/2*, *orf768a*) were screened out for the transmembrane domain (Appendix A, Appendix A). Among these ORFs, *orf768a* was predicted to be due to mutations in *cox1* that resulted in a stop codon. As shown in Appendix A, nine base mutations were found in *orf768a* by comparing the sequences of *cox1* and o*rf768a*. Among these SNPs, one caused a synonymous mutation, and three caused non-synonymous mutations. In addition, the last SNP (T–G) located on 1495 bp resulted in the continuation of the transcription of *cox1* and in a longer orf—*orf768a*, and the additional 810 bp of *orf768a* were located in the unique region of W99A (Figure 5a). The predicted protein product of *orf768a* possessed 11 transmembrane domains (Figure 5b), and the tertiary structures of the two proteins were very different (Figure 5c,d). Therefore, we hypothesized that *orf768a* was the strongest candidate for CMS celery.

### 2.8. Transcriptional Detection of CMS-Associated Candidate ORFs and Development a CMS Marker

To determine whether these ORFs were specific in the CMS celery, we confirmed by RT-PCR analysis, as shown in Figure 6a, that six ORFs (*orf115b*, *orf163a*, *orf254a*, *orf340a-1/2*, *orf402a-1/2*, *and orf768a*) were specific to the CMS celery, and *orf241a* was amplified neither in W99A nor in W99B.

Distinguishing the CMS seedlings in the offspring could reduce the cost of seed production when using the CMS line as a female parent in F_1_ seed production. In this study, we developed a CMS marker based on the six ORFs (*orf115b*, *orf163a*, *orf254a*, *orf340a-1/2*, *orf402a-1/2*, *orf768a*), and tested 41 celery varieties with a male-sterile phenotype (Appendix A). The results showed that only the orf768a marker was co-segregated with the CMS trait (Appendix A and Figure 6b), indicating that this marker can be used to screen CMS in celery.

## 3. Discussion

In this study, we first report the complete mitochondrial genome sequence of celery, which could be a valuable resource for future studies on the evolution of mitochondrial genomes in Apiaceae.

CMS is very important for hybrid seed production, especially for celery, which uses cross-pollination. Its flower organs are small and numerous, the flowering period is long and scattered, and the maturation stage of male and female flowers is inconsistent. Because of these characteristics, manual emasculation is extremely difficult, and the purity of hybrid seeds cannot be guaranteed. Therefore, using CMS can improve the utilisation of celery heterosis and promote the progress of celery cross-breeding.

Thus far, it is well accepted that CMS can be caused by SNPs and InDels. In addition, the rearrangement and recombination in the mitochondrial genome can result in new ORFs, which can also cause CMS. These ORFs often have chimeric structures, co-transcribed with known mitochondrial genes, and their encoded proteins often have transmembrane domains [3,4]. In this study, by comparison of the mitochondrial genomes of the celery CMS line W99A and its maintainer line W99B, 21 specific ORFs were identified in W99A. Among these 21 specific ORFs, there were seven ORFs (*orf115b*, *orf163a*, *orf241a*, *orf254a*, *orf340a-1/2*, *orf402a-1/2*, *orf768a*) harbouring transmembrane domains in their encoding proteins. Regarding the chimeric structure and co-transcription with functional genes, only *orf768a* harboured these two features; its 5′ end contained the almost complete sequence of the *cox1* gene, and 3′ end contained an unidentified sequence (Figure 5a). Therefore, it was considered a strong candidate gene for CMS. A similar phenomenon was observed in other CMS plants, such as the CMS genes in onion, radish, and rice, all of which have a chimeric structure with *cox1* [8,10,11,14,15,16,20].

As we know, in the production of hybrid F_1_ seeds, the restorer line and CMS line are usually used as male and female parents. However, no restorer line has been found or reported for celery until now. In addition, Apium flowers, which are tiny and numerous, and form umbels, are very easy to self-pollinate, so the most efficient emasculation method for celery is the use of sterile male lines. For celery, the CMS lines could be used as the female parent and any other fertile varieties as the male parent when producing hybrid F_1_ seeds, as the main edible part of celery is petiole and leaf blade, and the sterility of F_1_ hybrid seeds does not affect the yield of the celery hybrid. Up to date, reports on sterile male sources in celery are very few. The Iranian accession ‘P1229526’ was the first reported celery male sterile line, and it was controlled by a single recessive gene-- *ms-1* [29]. Besides that, the Chinese researchers Gao et al. reported a celery male sterile line ‘01-3A’ in an inbred line ‘01-3’, both of which are GMS [30]. Gao et al. selected the CMS line 0863A by crossing ‘01-3A’ and two other celery varieties [31]. However, the cause of CMS in celery has not yet been reported.

This study first reported the celery CMS line W99A, identified a strong candidate CMS gene, *orf768a*, and developed a CMS marker based on this gene, which is co-segregated with the CMS trait in 41 celery varieties (Appendix A; Figure 6b). The results contribute to understanding the cause of CMS formation in celery and promote the cross-breeding of celery and improve breeding efficiency.

## 4. Materials and Methods

### 4.1. Plant Materials

In 2005, a male sterile plant was found in a high generation inbred line population of the Chinese celery “tanzhixiangqin”. The stamens of this male-sterile plant were completely degenerated; that is, there were no stamens at flowering. Then taking this sterile plant as female parent and several Chinese and Western celery varieties as male parents to carry out paired test crosses and continuous backcrosses, a number of celery male sterile lines and maintainer lines with excellent comprehensive characters were selected. The celery CMS line W99A and its maintainer line W99B used in this study were one of the above-mentioned male sterile lines and maintainers, and W99A and W99B were backcrossed for more than 7 generations in our lab. Approximately 50 seeds of each line were sown in a greenhouse at 28 °C, 70% humidity, 16 h light/8 h darkness. Tender and white roots were collected at the four-leaf stage, immediately frozen in liquid nitrogen, and stored at −80 °C for mitochondrial DNA extraction and other uses.

### 4.2. Paraffin Sectioning and Microscopy

Flower buds from celery CMS line W99A and its maintainer line W99B were collected at six different developmental stages and immediately preserved in fresh FAA solution. Paraffin sections were then prepared according to the method of Cheng et al. [32].

### 4.3. Mitochondrial DNA Extraction and Quality Assess

Celery mitochondrial DNA was extracted by differential centrifugation. Approximately 100 g celery tender roots were ground with 1 L buffer A (0.4 mol/L mannitol, 50 mmol/L Tris-HCl, 1 mmol/L Na_2_EDTA, 5 mmol/L KCl, Ph = 7.5; Add 2 mmol/L β-Mercaptoethanol, 0.1% BSA, 10 mg mL polyvinylpyrrolidone before use), filtered with gauze to collect the filtrate. The filtrate was centrifuged at 3000 rpm for 15 min at 4 °C. The supernatant was collected, centrifuged at 4 °C, 10,000 rpm for 25 min, and the supernatant was discarded. The precipitates were suspended using buffer A (0.4 mol/L mannitol, 50 mmol/L Tris-HCl, 1 mmol/L Na_2_EDTA, 5 mmol/L KCl, pH = 7.5), and then centrifuged at 10,000 rpm, for 25 min at 4 °C, for collecting the precipitates. The precipitates were resuspended again using buffer A to produce the coarse mitochondria sample.

MgCl_2_ was added to a final concentration of 5 mmol/L and DNase I to a final concentration of 30 μM. After 1 h in ice bath, Na_2_EDTA was added to a final concentration of 15 mmol/L to terminate the DNA enzymolysis reaction. Coarse mitochondria were placed on top of a discontinuous sucrose gradient (sucrose concentration was 20% 7 mL, 40% 10 mL, 52% 10 mL, 60% 7 mL, prepared by buffer C: 1 mmol/L Na_2_EDTA, 15 mmol/L Tris-HCl, pH = 7.2). Ultracentrifugation at 20,000 r/min, for 2.5 h at 4 °C, placed the mitochondria at the interface of 40% and 52% sucrose. The mitochondria fraction was diluted 4 times the volume with buffer B (0.2 mol/L mannitol, 10 mmol/L Tris-HCl, 1 mmol/L Na_2_EDTA, pH 7.2) and centrifuged at 10,000 r/min for 15 min at 4 °C. These precipitates were mitochondrial DNA, which is pure, intact, and free from nuclear DNA contamination.

The mitochondrial DNA was suspended in lysis buffer, and 1/10 volume of 10% SDS and 1/100 volume of 30 mg/mL RNase solution, and incubated in a water bath at 50 °C for 30 min. Then, a 1/150 volume of 25 mg/mL proteinase K was added, and the sample was incubated in a water bath at 37 °C for 30 min. Mitochondrial DNA was extracted using phenol, phenol: chloroform: isoamyl alcohol (25:24:1), and chloroform. Afterward, 1/10 volume of 3 mol/L sodium acetate and 2 times volume of absolute ethanol were added, mixed well, stored at −20 °C for at least 30 min, and then centrifuged at 4 °C and 13,000 rpm for 15 min to collect mitochondrial DNA, which was washed with 75% ethanol–2–3 times. Then, mitochondrial DNA was precipitated, airdried, and dissolved in a small amount of TE buffer.

The purity and quality of mitochondrial DNA were assessed using Qubit 3.0 and NanoDrop2000.

### 4.4. Mitochondria DNA Sequence and Genome Assembly

Purified mitochondrial DNA (1 μg) was fragmented for 300–500-bp paired-end library construction using the NEBNext Ultra DNA Library Prep Kit. Sequencing was performed on an Illumina NovaSeq 6000 platform (BIOZERON Co., Ltd., Shanghai, China). Approximately 6.2 Gb and 5.9 Gb of raw data from W99A and W99B were generated using 150 bp paired-end read lengths, respectively. For PacBio library construction, approximately 5 μg of sheared and concentrated DNA were used for size selection by Blue Pippin. Approximately 20 kb SMRTbell libraries were constructed according to the manufacturer’s instructions (PacBio, Menlo Park, CA, USA). A total of 73.6 Gb (W99A) and 76.9 Gb (W99B) data were obtained using the PacBio Sequel platform ((PacBio, Menlo Park, CA, USA)).

For the celery mitochondria genome assembly, we first used the carrot mitochondrial genome (JQ248574) as the reference mitochondrial genome by using NOVOPlasty [33] for de novo assembly, and we extracted a number of potential mitochondrion reads from the pool of Illumina reads by using BLAST searches against mitochondrial genomes of related species carrot and the NOVOPlasty result. We then used the above Illumina reads to perform the mitochondrion genome using the SPAdes-3.13.0 package [34]. Further, we used the BWA men program to align the clean PacBio long reads against the NOVOPlasty and SPAdes assembled scaffolds. In addition, all of the aligned PacBio reads underwent self-correction. We used the Canu v2.1.1 package [35] and error correction for the mitochondrial genome de novo assembly. Finally, the assembled PacBio sequences were checked for overlaps and were connected with each other.

### 4.5. Sequence Analysis and Genome Annotation

The predicted protein-coding genes, tRNA and rRNA, were annotated using the online GeSeq tool [36]. We used OGDRAW v1.2 [37] to draw a circular map of the mitochondrial genome.

### 4.6. RT-PCR of CMS-Associated Candidate ORFs

Total RNA was isolated from W99A and W99B buds using an RNA extraction kit (SV Total RNA Isolation System, Promega, Madison, WI, USA). First-strand cDNA was synthesised using PrimeScript™ RT Kit (Takara, RR037A, Kusatsu, Shiga, Japan). RT-PCR was performed using the primers listed in Appendix A, with the following amplification conditions: 94 °C for 5 min; 30 cycles of 98 °C for 10 s, 55 °C for 15 s, and 72 °C for 2 min; and a final 10 min extension at 72 °C.

## Figures and Tables

**Figure 1 ijms-22-08584-f001:**
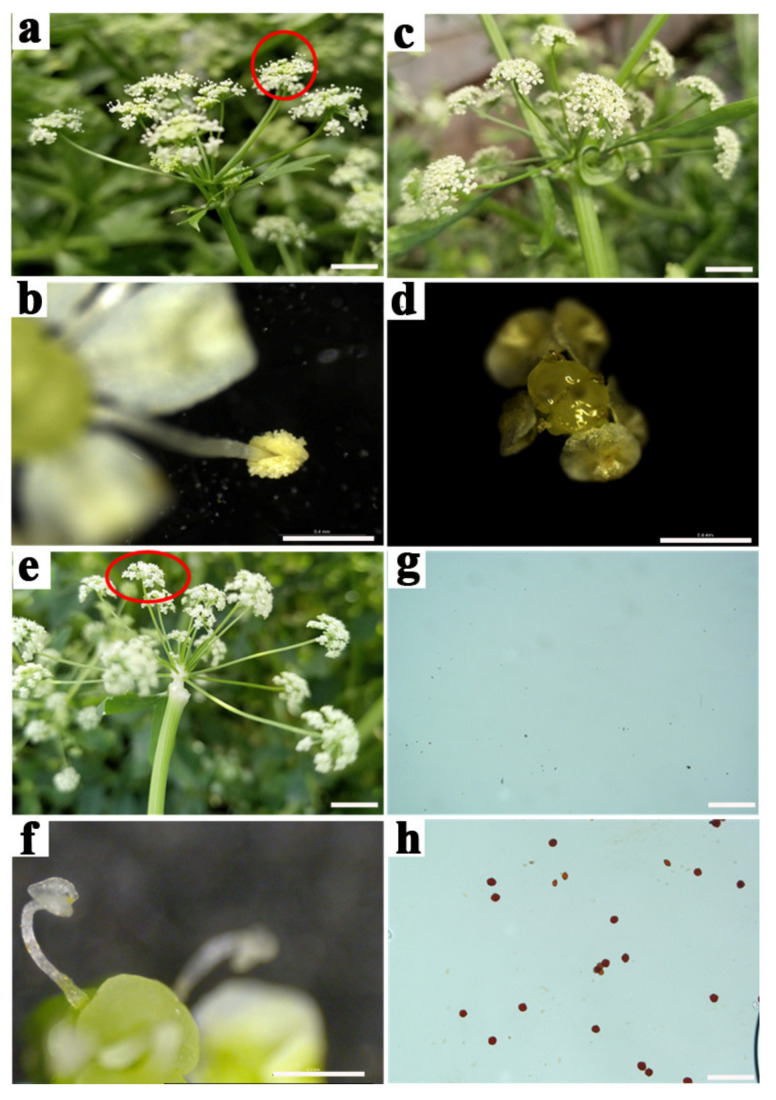
Phenotype of flowers from the celery CMS line W99A and its maintainer line W99B. (**a**) Flower of maintainer line W99B; (**b**) anther of maintainer line W99B; (**c**) flower of completely stamen degeneration type; (**d**) anther of completely stamen degeneration type; (**e**) flower of incomplete stamen degeneration type; (**f**) anther of incomplete stamen degeneration type; (**g**,**h**) TTC staining for pollen viability of the celery CMS line W99A and its maintainer line W99B. (**a**,**c**,**e**) scale = 5 cm; (**b**,**f**) scale = 0.4 mm; (**d**) scale = 0.8 mm; (**g**,**h**) = 200 µm.

**Figure 2 ijms-22-08584-f002:**
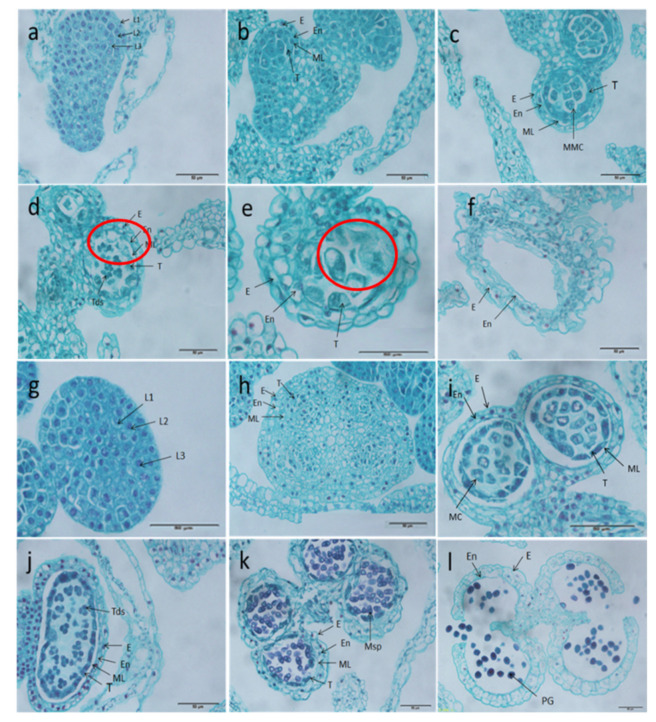
Microspore development of anthers from the celery CMS line W99A and its maintainer line W99B. (**a**–**f**) Celery CMS line W99A; (**g**–**l**) maintainer line W99B. (**a**,**g**) stamen primordium stage; (**b**,**h**) sporogonia and sporogenetic cell stage; (**c**,**i**) pollen mother cell stage; (**d**,**j**) tetrad stage, (**e**,**k**), uninucleate stage; (**f**,**l**) pollen mature stage. MMC: microspore mother cell, Msp: microspore, MC: meiotic cells, T: tapetum, Tds: tetrad microspore, PG: pollen grain. E: Extexine, En: endodermis, ML: middle lamella. Scale bars = 50 µm.

**Figure 3 ijms-22-08584-f003:**
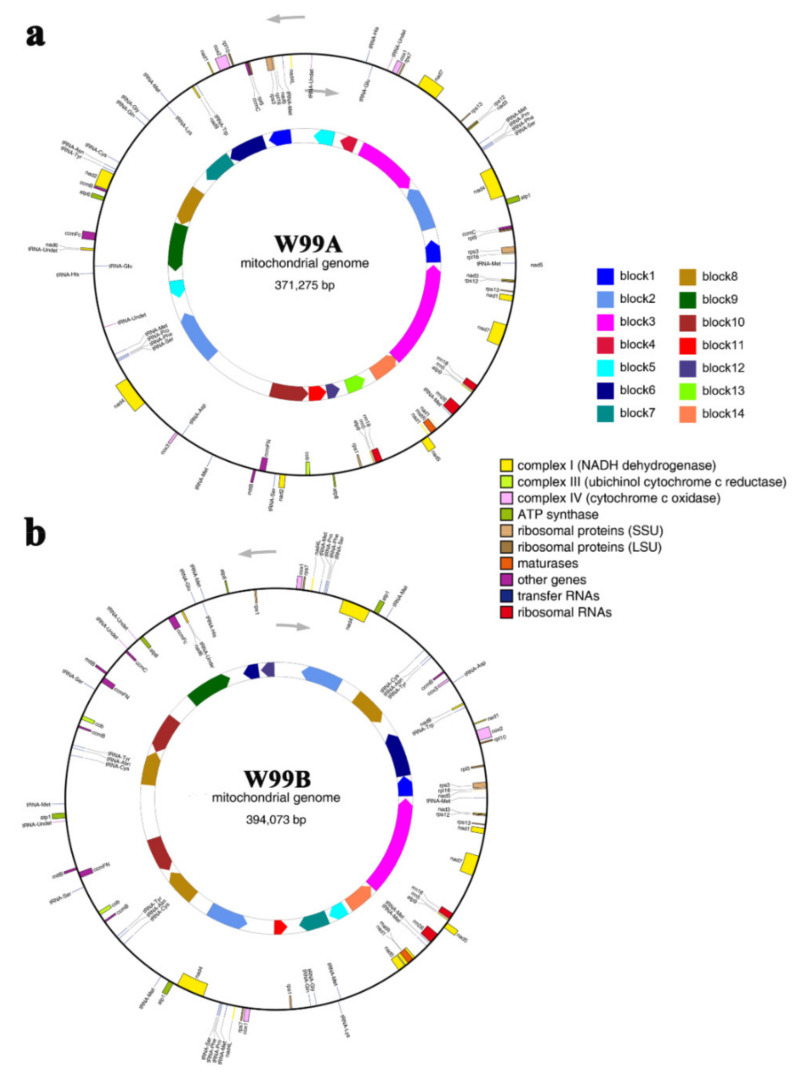
Mitochondrial genome maps of the celery CMS line W99A (**a**) and its maintainer line W99B (**b**). Genes names inside and outside of the circle indicate clockwise and counterclockwise transcription, respectively. Different colours indicate different functions of gene products. The inner circles indicates syntenic sequence blocks.

**Figure 4 ijms-22-08584-f004:**
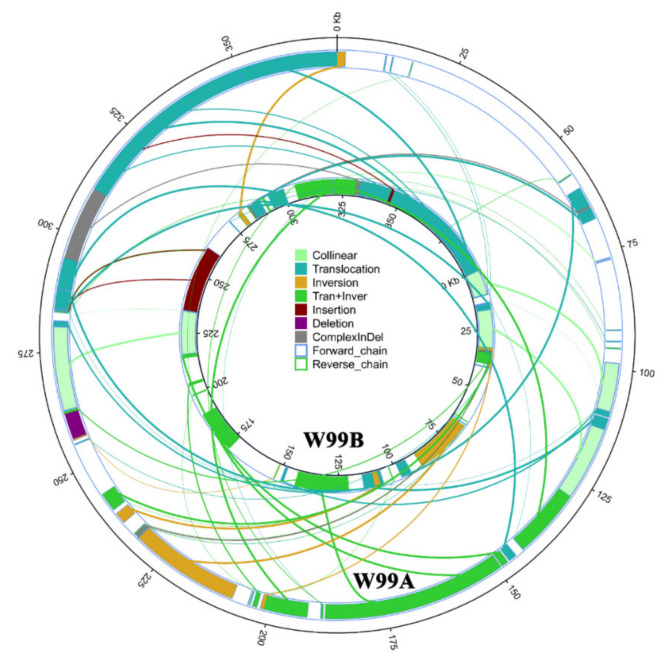
The structural variation map of W99A and W99B mitochondrial genomes.

**Figure 5 ijms-22-08584-f005:**
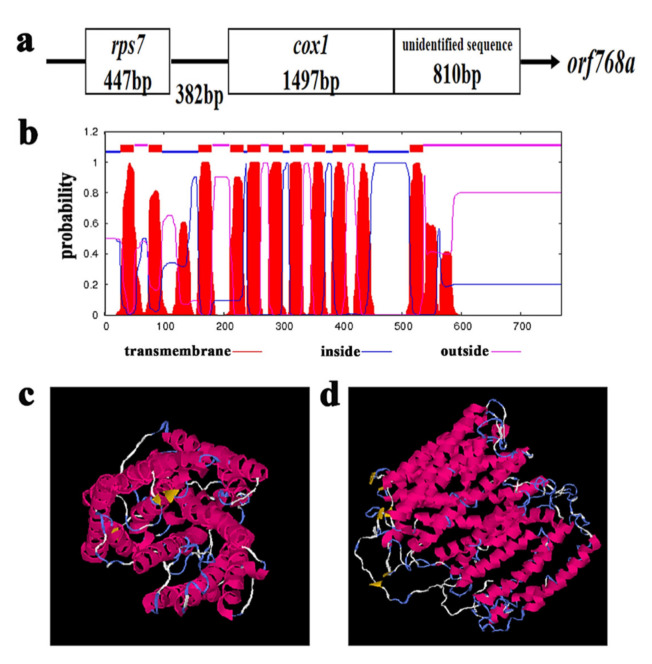
Gene structure of *orf768a*. (**a**) The *orf768a* sequence has a chimeric structure including *cox1,* arrow indicates the 5′-to-3′ direction. (**b**) Transmembrane domains of the *orf768a* gene products. (**c**) Tertiary structure analysis of *cox1*. (**d**) Tertiary structure analysis of *orf768a*.

**Figure 6 ijms-22-08584-f006:**
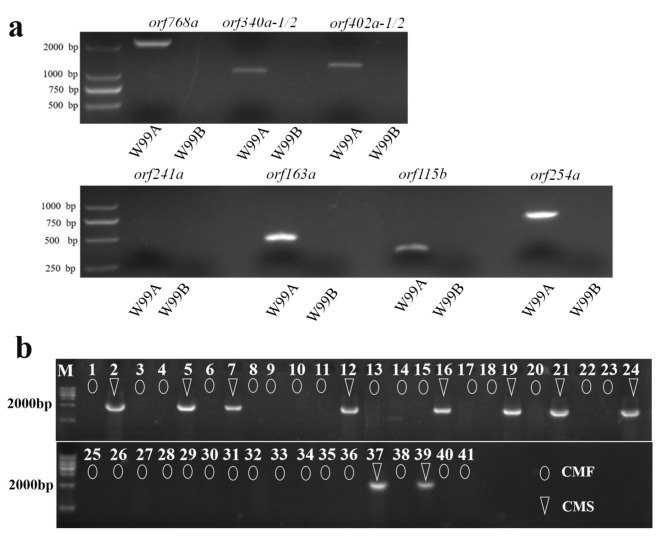
PCR confirmation of the specificity of ORFs for CMS celery. (**a**) RT-PCR detection of *orf768a*, *orf340a-1/2*, *orf340a-1/2*, *orf241a*, *orf115b*, *orf163a*, *orf254a*; (**b**) CMS maker orf768a tests in 41 pepper inbred lines.

## Data Availability

The complete mitochondrial genome sequences of W99A (MK562755) and W99B (MK562756) can be found and downloaded in NCBI (https://www.ncbi.nlm.nih.gov/genbank/).

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
