# Peer review of "Complete Mitochondrial Genome Sequence and Identification of a Candidate Gene Responsible for Cytoplasmic Male Sterility in Celery (Apium graveolens L.)"

_ijms, 2021, doi:10.3390/ijms22168584_

Round 1
Reviewer 1 Report
In the submitted manuscript “Complete mitochondrial genome sequence and identification 2 of a candidate gene responsible for cytoplasmic male sterility 3 in celery (Apium graveolens L.)” Cheng et al. identified 15 ORFs from 21 unique regions by performing a comparative analysis of the complete mitochondrial genome sequences between the celery CMS line W99A and its maintainer line W99B, they further confirmed that unique chimeric gene, orf768a is a candidate gene for CMS in W99A by protein domain prediction analysis, and they also developed a co-segregate marker for screening the CMS in celery. Overall, this is a good report with the experiments reasonably arranged to convey the key messages. However, the conclusions will be more convincing if the authors can provide some experimental data from functional analysis of orf768a such as overexpression, mutagenesis and localization assay.
Minor point:
- The manuscript writing can be further improved especially in grammar, I made several edits in Abstract for the authors to consider in their revised manuscript in the attached file.
- Add a scale bar in Figure 1a-g and h.
- Need to provide a reference gene for RT-PCR analysis.

Author Response
Reviewer 1
In the submitted manuscript “Complete mitochondrial genome sequence and identification 2 of a candidate gene responsible for cytoplasmic male sterility 3 in celery (Apium graveolens L.)” Cheng et al. identified 15 ORFs from 21 unique regions by performing a comparative analysis of the complete mitochondrial genome sequences between the celery CMS line W99A and its maintainer line W99B, they further confirmed that unique chimeric gene, orf768a is a candidate gene for CMS in W99A by protein domain prediction analysis, and they also developed a co-segregate marker for screening the CMS in celery. Overall, this is a good report with the experiments reasonably arranged to convey the key messages. However, the conclusions will be more convincing if the authors can provide some experimental data from functional analysis of orf768a such as overexpression, mutagenesis and localization assay.
Answer: Thank you for constructive comments. As the genetic transformation system of celery is very difficult, we have not carried out the relevant functional analysis of orf768a, but we are exploring the establishment of the genetic transformation system of celery and will verifying the function of orf768a in future research.
Minor point:
- The manuscript writing can be further improved especially in grammar, I made several edits in Abstract for the authors to consider in their revised manuscript in the attached file.
Answer: Thank you for constructive comments, we had carefully revised the manuscript according to the reviewers' comments, and also have re-scrutinized to improve the English by a language polishing service (Editage (www.editage.cn), please see the Editing certificate and the file (Manuscript-editing-track-7.5). Besides that, we also uploaded a clean manuscript, which we had carefully modified according the English editing and all the comments from the reviewers (file name: Manuscript-clean-2021.7.5), as a convenient for editors and reviewers to review again.
- Add a scale bar in Figure 1a-g and h.
Answer: we had added a scale bar in Figure 1a-g and h, see Figure 1.
- Need to provide a reference gene for RT-PCR analysis.
Answer: The reference gene for RT-PCR analysis was shown in Supplementary Table S12, and meanwhile we have added the related information the manuscript (file name: Manuscript-clean-2021.7.5) in to the M&M part (line: 285).
Reviewer 2 Report
The manuscript "Complete mitochondrial genome sequence and identification of a candidate gene responsible for cytoplasmic male sterility in celery (Apium graveolens L.)" describes the complete mitogenome of celery, which has the size and structure typical for angiosperm mitochondrial genomes. the authors also found the putative CMS gene orf768a, based on the comparison of the mitogenomes between the male sterile and maintainer lines, which is of importance for agricultural use.
I have two major comments.
- The annotation of the repeats of various size in the mitogenome of celery would increase the value of the study for understanding to the evolution of the plant mitogenome structure.
- The comparison of the candidate CMS gene expression between the male sterile plants and their siblings with restored male fertility will bring another evidence for the CMS function. Please, provide this comparison.
The extensive correction of the text by a native English speaker is required.
Author Response
Reviewer 2
Comments and Suggestions for Authors
The manuscript "Complete mitochondrial genome sequence and identification of a candidate gene responsible for cytoplasmic male sterility in celery (Apium graveolens L.)" describes the complete mitogenome of celery, which has the size and structure typical for angiosperm mitochondrial genomes. the authors also found the putative CMS gene orf768a, based on the comparison of the mitogenomes between the male sterile and maintainer lines, which is of importance for agricultural use.
I have two major comments.
- The annotation of the repeats of various size in the mitogenome of celery would increase the value of the study for understanding to the evolution of the plant mitogenome structure.
Answer: Thank you for affirming our work and for your constructive comments.
- The comparison of the candidate CMS gene expression between the male sterile plants and their siblings with restored male fertility will bring another evidence for the CMS Please, provide this comparison.
Answer: We had conducted the candidate CMS gene (orf115b, orf163a, orf254a, orf340a-1/2, orf402a-1/2, and orf768a) expression in this revision, and had added the related information in line 173-175 (file name: Manuscript-clean-2021.7.5) and Supplementary Figure S2.
The extensive correction of the text by a native English speaker is required.
Answer: Thank you for constructive comments, we had carefully revised the manuscript, and also have re-scrutinized to improve the English by a language polishing service (Editage (www.editage.cn), please see the Editing certificate and the file (Manuscript-editing-track-7.5). Besides that, we also uploaded a clean manuscript, which we had carefully modified according the English editing and all the comments from the reviewers (file name: Manuscript-clean-2021.7.5), as a convenient for editors and reviewers to review again.
Reviewer 3 Report
I would recommend the authors to replace figure 4 with the highest resolution (quality) image.
Author Response
Reviewer 3
Comments and Suggestions for Authors
I would recommend the authors to replace figure 4 with the highest resolution (quality) image.
Answer: Thank you for constructive comments, we had replaced Figure 4 with the highest resolution in this revision. Please see Figure 4.
Round 2
Reviewer 1 Report
This revised version of the manuscript improved in terms of addressed my prior comments, I would be happy to recommend for acceptance in light of changes made in response to following comments:
To obtain accurate and reliable results in RT-PCR analyses, the authors should add a corresponding gel image of reference gene (AgACTIN) expression in Figure 6a for evaluation of transcript level.
Author Response
This revised version of the manuscript improved in terms of addressed my prior comments, I would be happy to recommend for acceptance in light of changes made in response to following comments:
To obtain accurate and reliable results in RT-PCR analyses, the authors should add a corresponding gel image of reference gene (AgACTIN) expression in Figure 6a for evaluation of transcript level.
Answer:Thank you for your constructive comments. In this study, these candidate CMS genes are not detected in W99B (Figure 6a), which means these candidate CMS genes do not exist in W99B, so it no necessary to do the qPCR for measuring the expression of these candidate CMS genes. So in this revision, we had deleted the related information, and there is no need to add the corresponding gel image of reference gene (AgACTIN) expression in Figure 6a.
Reviewer 2 Report
The authors of the revised manuscript did not follow my comments.
The repeats were not marked and described in the mitogenome - at least I did not discover it.
RT qPCR to measure the expression of CMS genes was not performed correctly. First, the siblings - female and restored hermaphrodites shall be compared, both sharing excatly the same mitogenome. W99A and W99B possess distinct mitogenoems.
Second, mitochondrial reference gene shall be used as a reference, not nuclear-encoded gene.
Author Response
Comments and Suggestions for Authors
The authors of the revised manuscript did not follow my comments.
The repeats were not marked and described in the mitogenome - at least I did not discover it.
Answer: Thank you for your constructive comments. We had added the repeats analysis in this revision, please see line 126--132 and Supplementary Table S4.
RT qPCR to measure the expression of CMS genes was not performed correctly. First, the siblings - female and restored hermaphrodites shall be compared, both sharing excatly the same mitogenome. W99A and W99B possess distinct mitogenoems.
Second, mitochondrial reference gene shall be used as a reference, not nuclear-encoded gene.
Answer: Thank you for your constructive comments. In this study, these candidate CMS genes are not detected in W99B (Figure 6a), which means these candidate CMS genes do not exist in W99B, so it is no necessary to do the qPCR for measuring the expression of these candidate CMS genes. So in this revision, we had deleted the related information.
Round 3
Reviewer 2 Report
The authors of "Complete mitochondrial genome sequence and identification of a candidate gene responsible for cytoplasmic male sterility in celery (Apium graveolens L.)" improved their manuscript. But one issue persists. Please compare the CMS candidate gene expression in females and hermaphrodites from the same cross, carrying the same mitogenome. If the CMS gene cannot be restored in celery, please indicate and explain this fact !!